

# Semiannual variation of Pc5 ULF waves and relativistic electrons over two solar cycles of observations: comparison with predictions of the classical hypotheses

Facundo L. Poblet[1], Francisco Azpilicueta[1], and Hing-Lan Lam[2]

[1]CONICET, Facultad de Cs. Astronómicas y Geofísicas, Universidad Nacional de La Plata, La Plata, Buenos Aires, Argentina.
[2]Geomagnetic Laboratory, Natural Resources Canada, Ottawa, Ontario, Canada.

**Correspondence:** Facundo L. Poblet (fpoblet@fcaglp.unlp.edu.ar)

**Abstract.** Pc5 ULF (ultra-low frequency) waves can energize electrons to relativistic energies of >2MeV in geostationary orbits. Enhanced fluxes of such electrons can induce operational anomalies in geostationary satellites. The variations of the two quantities in time scales ranging from days to solar cycles are thus of interest in gauging their space weather effects over different time frames. In this study, we present a statistical analysis of two 11-year solar cycles (Cycle 22 and 23) of

data comprising the daily relativistic electron fluence observed by GOES geostationary satellites and daily Pc5 power derived from auroral zone magnetic observatories in Canada. Firstly, an autocorrelation analysis is carried out, which indicates 27-day periodicity in both parameters for all solar phases, and such a periodicity is most pronounced in the declining and late-declining phase. Also, a 9-day and 13-day periodicity, though not present in all the years, are seen in some years. Then, a superposed epoch analysis is performed to scrutinize Semiannual Variation (SAV), which shows fluence near the equinoxes is one order of

magnitude higher than near solstices and Pc5 power is 0.5 orders of magnitude higher near the equinoxes than near the solstices. We then evaluate three possible SAV mechanisms (which are based on the Axial, Equinoctial, and Russel & McPherron effect) to determine which one can best explain the observations. Correlation of the profiles of the observational curves with those of the angles that control each of the SAV mechanisms suggests that the Equinoctial mechanism may be responsible for the SAV of electron fluence while both the Equinoctial and the Russell & McPherron mechanisms are important for the SAV of Pc5 power. Comparable results are obtained when using functional dependencies of the main angles instead of the angles

mentioned above. Lastly, superposed curves of fluence and Pc5 power were used to calculate least-square fits with a fixed semiannual period. Comparison of maxima and minima of the fits with those predicted by the three mechanisms shows that the Equinoctial effect better estimates the maxima and minima of the SAV in fluence while for the SAV in Pc5 power the Equinoctial and Russell & McPherron mechanisms predict one maximum and one minimum each.

# 1 Introduction

Relativistic electrons with energies >2 MeV can penetrate the surface of a satellite and cause internal charging that can induce satellite operational anomalies, as conclusively demonstrated by Wrenn (1995). Internal charging by relativistic electrons not





only causes satellite operational anomalies that are a nuisance to satellite operators, but can also render the complete failure of a satellite, as exemplified by the consecutive outages of Telesat Canada's Anik-E1 and E-2 geostationary satellites on 20 January 1994 that wreaked havoc in communication across Canada for hours (Baker et al., 1994a, b; Lam et al., 2012). There are other serious satellite incidents due to internal charging by relativistic electrons such as the Anik-E1 failure on 26 March

1996 (Baker et al., 1996). The intensification of relativistic electrons that can cause satellite problems have been shown to be associated with Pc5 ULF (ultra-low frequency) waves (Rostoker et al., 1998; Mathie and Mann, 2001; Mann et al., 2004; Simms et al., 2014; Lam, 2017). The acceleration mechanisms of relativistic electrons attributable to Pc5 ULF waves can be due to magnetic pumping (Borovsky, 1986; Liu et al., 1999), drift-resonant acceleration (Elkington et al., 1999), transit-time acceleration (Summers and yu Ma, 2000), and the popular radial diffusion (e.g., Falthammar, 1968; Schulz and Lanzerotti,

1974; Perry, 2005; Ozeke et al., 2014). No matter what the actual acceleration mechanism or process is, Lam (2017) has shown that Pc5 power has the potential of predicting relativistic electrons that can harm satellites. It is, therefore, pertinent to peruse the time variations of Pc5 ULF waves and relativistic electrons together in detail in order to appraise their space weather effects over different time assemblies.

In this work we analyze both ground-based Pc5 magnetic pulsations, which are a manifestation of Pc5 ULF waves, and

relativistic electrons at geostationary orbit, focusing on their time variations from a few days to a Solar Cycle (SC). An extended analysis is carried out for a particular kind of variation known as the Semiannual Variation (SAV). SAV is an annual phenomenon, characterized by maximum levels of activity near equinoxes and minima near solstices and it can be detected in a diverse set of solar-terrestrial measurements (Azpilicueta and Brunini, 2011, 2012; Vichare et al., 2017; Bai et al., 2018), including relativistic electrons of the outer Van Allen belt (Baker et al., 1999; Li et al., 2001; Kanekal et al., 2001) and ULF waves

(Sanny et al., 2007; Rao and Gupta, 1978). In the first case Baker et al. (1999) used measurements of both the low-altitude SAMPEX and high-altitude POLAR spacecraft to calculate quarterly averages centered at the equinoxes and solstices. They found that the fluxes were nearly three times higher at the equinoxes than at solstices which means a semiannual modulation in these measurements (McPherron et al., 2009). Moreover, SAMPEX observations were also used by Kanekal et al. (2010) to study the dependence of the SAV in relativistic electrons with a wide range of L-shells covering the descending and ascending

parts of a SC. Their results showed that the flux peaks were delayed about 30 days from the times of the nominal equinoxes during the descending phase. But in the ascending phase, the lag times were asymmetrical for both equinoxes.

In the case of ULF waves, Sanny et al. (2007) examined the seasonal and diurnal pattern of ULF wave powers, using magnetic measurements from Geostationary Environment Satellites (GOES) sensors. They studied Pc3, Pc4 and Pc5 pulsations, which all clearly exhibit the June/July minimum. They also identified a strong local minimum in Pc4 band power around noon,

whereas the minima of the Pc5 and Pc3 bands appeared to be distributed on the dayside. All the frequency bands had elevated power levels around local midnight. An older work where a SAV is reported in Pc5 pulsations was published by Rao and Gupta (1978). They found the SAV to be particularly evident in the morning hours, close to $8 \pm 1$ h LT.

There are three mechanisms that are commonly referred to in the literature to explain the SAV and each one seems to be controlled by an angle. The first mechanism is known as the Axial hypothesis and the angle is the Earth's heliographic latitude.

This angle reaches maximum absolute values about 14 days before the equinoxes (see Table 5) when the Earth approaches





high-speed solar wind regime such as sunspot region (Cortie, 1912) or coronal holes. The high solar wind speed originating from these regions might be the driver of the enhancements in the activity. On the contrary, the Earth crosses regions of slow-speed solar wind approaching the solstices, at the proximity of the Sun's equator and then there is minimum activity (Phillips et al., 1995).

The second mechanism is known as the Russell & McPherron (RM) hypothesis (Russell and McPherron, 1973), which establishes that there is a varying probability of a southward directed component of the Interplanetary Magnetic Field throughout the year. This leads to different probability of magnetic reconnection between the Interplanetary Magnetic Field and the terrestrial magnetic field lines at the nose of the magnetopause. Near the equinoxes(solstices) the probability is maximum(minimum). The relevant angle is the angle between $z^{GSM}$ and $y^{GSEq}$ (GSM: Geocentric Solar Magnetospheric, GSEq: Geocentric Solar
Equatorial coordinate system).

       The last mechanism is known as the Equinoctial hypothesis (Bartels, 1932). Boller and Stolov (1970) showed that in theory, the Kelvin-Helmholtz instability originated by the viscous-like interactions between the solar wind and the magnetosphere along the flanks of the magnetosphere, predicts a semiannual pattern with instability maxima(minima) near the equinoxes(solstices). This is thought to be the physical process behind the Equinoctial theory. The controller angle is the one
delimited by the SW direction and the Earth's dipole.

       A main objective of this work is to test which one of these mechanisms better predicts the SAV that we find in Pc5 pulsations and in relativistic electrons. The procedure involves the comparison between observational curves and the shape of the relevant angles of each mechanism. This method has been applied before to look for the dominant mechanism in the geomagnetic activity (Roosen, 1966; Cliver et al., 2002) finding that the Equinoctial and RM effects are the dominant ones and the Axial
effect is the least important. This paper not only extends their work in magnetic activity in terms of Pc5 magnetic pulsations but also includes relativistic electrons in geostationary orbits. The consolidation of the two quantities in a single study on their semiannual variations and other periodicities over two solar cycles elucidates their space weather effects under different temporal contexts.

## 2   Data

### 2.1   GOES Relativistic electrons

As internal charging by relativistic electrons on satellites located at geosynchronous orbit is a function of integrated flux over time period, we use daily fluence values, which is an accumulation of fluxes over 24 hours, to represent the electron variations in this work. Specifically, we analyzed fluences of relativistic electrons with energies >2 MeV derived from flux measurements onboard NOAA's GOES. GOES are in geostationary orbit about 35790 km above Earth's surface in the equatorial plane at 6.6
$R_E$.

       The data span SCs 22 and 23 from June 1987 to December 2009. The same data suite has been used previously to study Pc5 ULF waves and relativistic electrons by Lam (2017), which provides details on GOES as well as GOES data. Table 1 spells





| Satellite | Start to end date | Slot position |
|---|---|---|
| GOES 7 | June 1987 to February 1995 | GOES-East/West |
| GOES 8 | March 1995 to March 1996 and August 1998 to March 2003 | GOES-East |
| GOES 9 | April 1996 to July 1998 | GOES-West |
| GOES 11 | January–February 2008, December 2008, and January–December 2009 | GOES-West |
| GOES 12 | April 2003 to December 2007 and March–November 2008 | GOES-East |

**Table 1.** Satellite data used in this study.

out the specific GOES used in this work with their time coverage and East or West allocation. The rationale behind the choice of these satellite is given in (Lam, 2017).

## 2.2 Pc5 power

To study ULF waves in the Pc5 frequency band we generate a time series of daily Pc5 power values using Canadian geomag-
netic data collected by the Canadian Magnetic Observatory System (CANMOS) (Lam, 2011). The geomagnetic data cover the same period as the electron data described in Section 2.1. The hourly values of Pc5 power derived using the $X$ component (northward component) of the Earth's geomagnetic field recorded at 1 min intervals were used in this study. 24 hourly powers of a UT day were added to obtain the daily power. In (Lam, 2017), daily power was the mean of the hourly powers in a UT day after the hourly powers in the midnight sector were excluded to avoid contamination of substorms to "pure" Pc5 ULF waves.
However, in order to investigate the semiannual variations fully in this study, the daily power includes contributions from the midnight sector so that all magnetic fluctuations in the Pc5 spectrum (Jacobs et al., 1964) in a UT day are considered. We refer the reader to (Lam, 2017) for more detail on the methodology used to obtain Pc5 power from the raw geomagnetic data.

The CANMOS observatories selected to calculate Pc5 power (see Table 2) are located in the Canadian auroral zone close to the footprints of magnetic field lines threading GOES in order to relate ground magnetic variations with relativistic electrons
near geostationary orbit. As can be seen from the last column of Table 2 the data come primarily from Fort Churchill station (FCC) that is located at a geographic longitude of 94.1° W which is approximately midway between GOES-East and GOES-West. Where there were gaps or spikes in FCC data, YKC data were used. When both FCC and YKC data were absent or not usable, PBQ data were used. When data from FCC, YKC, and PBQ were not available, data from BLC and CBB at the fringe of the auroral zone near the cusp were used to fill in the data gap. It can be seen from Table 2 that FCC and YKC together
cover ∼94 % of the total days processed with FCC contributing most of the data.

Many studies have been carried out using Pc5 power derived from a single magnetic station in the auroral oval as in this study (Glaβmeier, 1988; Trivedi et al., 1997; Mathie and Mann, 2001; Mann et al., 2004). As pointed out by Lam (2017), Kozyreva et al. (2007) noted that improvements made by global ULF wave index did not change the basic features of its temporal variations and that the results of the works of Mathie and Mann (2001) and Mann et al. (2004) obtained from nonglobal ULF





| Station | Code | Geographic latitude | Geographic longitude | Geomagnetic latitude | Geomagnetic longitude | Percentage over total days of measurements [%] |
|---|---|---|---|---|---|---|
| Fort Churchill | FCC | 58.8°N | 94.1°W | 68.8°N | 37.5°W | 82.90 |
| Yellowknife | YKC | 62.5°N | 114.5°W | 69.1°N | 67.3°W | 10.73 |
| Poste de-la-Baleine | PBQ | 55.3°N | 77.8°W | 66.8°N | 12.8°W | 4.00 |
| Baker Lake | BLC | 64.3°N | 96.0°W | 72.7°N | 35.5°W | 2.01 |
| Cambridge Bay | CBB | 69.1°N | 105.0°W | 76.2°N | 53.7°W | 0.37 |

**Table 2.** Coordinates of CAMMOS observatories used in this work.

wave power remain valid. It is therefore justified to use the Pc5 power from a single auroral zone station to generate a large data set for the statistical study in this work.

## 3  General characteristics of Pc5 power and fluence in SCs 22 and 23

Figure 1 presents an overview of the complete Pc5 power and fluence daily values plotted as black dots. This figure echoes
trends delineated in Figure 1 of (Lam, 2017), whose daily values exclude midnight sector contributions, as mentioned earlier.

The thick black lines are the 365-day moving average of the Pc5 power and fluence. The smoothed sequence of daily Sunspot Numbers has also been added (orange curve) to represent the SC which is useful when making reference to variations of the parameters to a specific SC phase.

The smoothed curves of Pc5 power and fluence can be used to highlight the underlying trends. For example, they indicate
high levels during the descending phases of both cycles. Differences in trends at different phases of a SC can also be seen. Although there appears to be minor variations in the trends between Pc5 power and fluence (e.g. Pc5 increasing while fluence decreasing in the early portion of SC 23 and Pc5 leveling while fluence depressing around SC 22 maximum) the gross features of their evolution in both SCs appear to be similar.

To see the relationship between Pc5 power and electron fluence, Figure 2 shows their variations in log for 1996 and 2007
during the lower portion of the descending phase of SC 22 and SC 23 respectively. The x axis corresponds to the Day Of Year (DOY). The peaks and valleys in Pc5 power and fluence seem to follow each other with a lag of about a couple of days in fluence peaks with respect to Pc5 power peaks. This time shift is clearer for 2007 than for 1996 and was studied in detail by Lam (2017), who concluded that Pc5 power can potentially be used to predict electron fluence 2 to 3 days in advance before the enhancements in electron fluence at geostationary orbit and also that the lag is smaller for extremely high fluence values.
Besides showing the relationship between Pc5 power and electron fluence, Figure 2 also indicates that in 1996, fluence values clearly demonstrate SAV pattern, which is not readily discernible at first glance when looking at other years. Furthermore, both years show regular variations in the two parameters. The SAV and the regular variations as exemplified here will be further investigated statistically in the sections below.





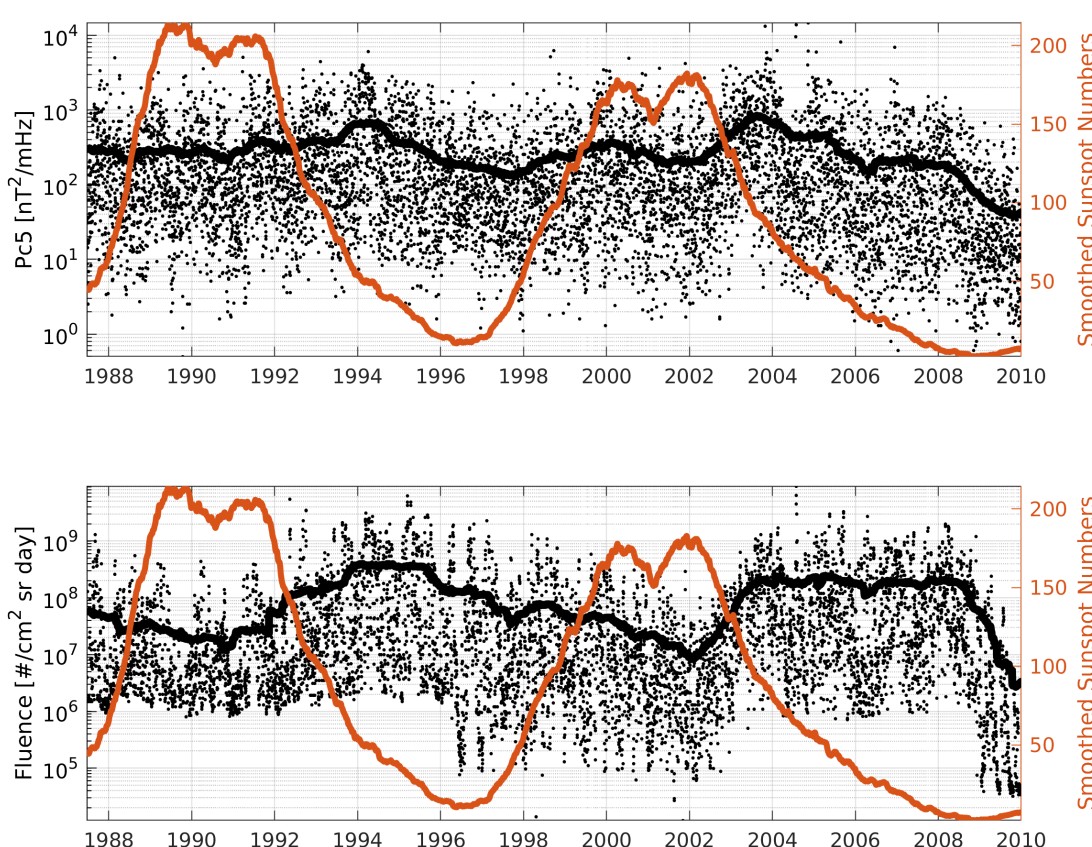

**Figure 1.** Upper panel: Pc5 power daily values represented as black dots. The thick black curve is the Pc5 power smoothed with a 365-day moving average. Bottom panel: Fluence daily values represented as black dots. The thick black curve is the fluence smoothed with a 365-day moving average. The orange curve corresponds to the yearly-smoothed Sunspot Numbers sequence.



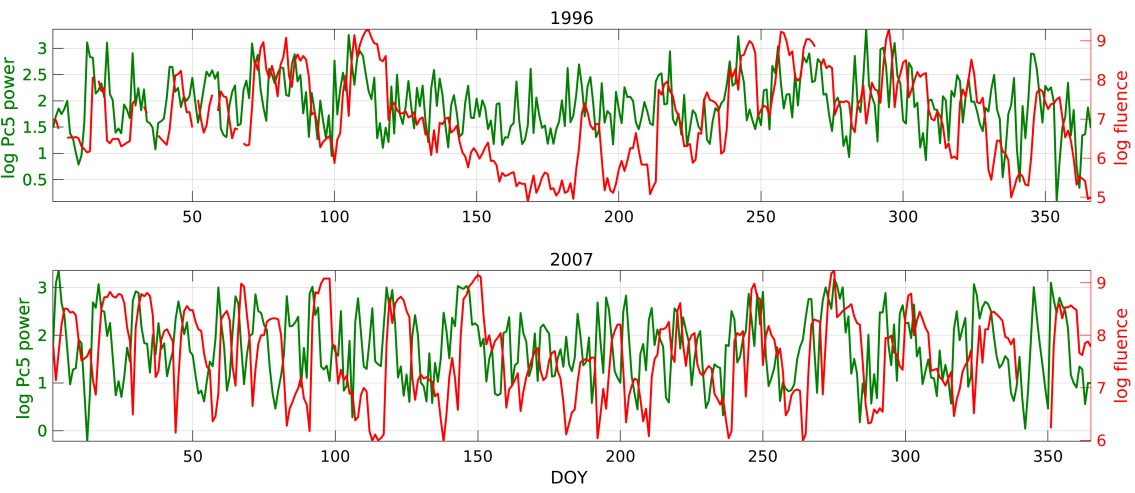

**Figure 2.** Fluence (red curve) and Pc5 power (green curve) values in 1996 (upper panel) and 2007 (bottom panel). The peaks and valleys of Pc5 power and fluence seem to follow each other closely, especially in 2007.

### 3.1 Dominant periodicities

#### 3.1.1 Autocorrelation functions

In order to investigate the dominant periodicities in Pc5 power and electron fluence, we calculated the autocorrelation function (ACF) of the logarithm of both parameters for specific years corresponding to different phases of a solar cycle. To establish

whether a value of correlation at a certain lag was significant or not, a criterion based on a Student's test (or "t-test") on the correlation coefficient $r$ was adopted. Following (Rodgers and Nicewander, 1988), the hypothesis of null correlation ($r = 0$) is rejected when $r$ satisfies:

$$|r| > \frac{t}{\sqrt{N - 2 + t^2}} \; ; \tag{1}$$

where $N$ is the length of sequence in days, and $t$ is the quantile of a Student's distribution (t-distribution) with $N - 2$ degrees of

freedom and a significance level of $1\%$. If the hypothesis can not be rejected, $r$ is statistically equivalent to zero and considered not significant.

Figures 3 and 4 present the ACFs of the logarithmic Pc5 power and fluence respectively for different phases of both SCs (22 and 23) as a function of lag in days. One representative year of the ascending, maximum, descending, late descending and minimum phases was selected based on the shape of the Sunspot Numbers in Figure 1. The years corresponding to each phase

are explicitly shown in the Figures.

Since $N$ remains constant for every lag value, there is a region in the plots inside which $r$ should be considered statistically null, as delimited by two Critical Values for Correlation (CVC). This region is bounded by two horizontal black lines located at $\pm$CVC. The obvious maximum value at lag 0 was excluded from the figures in order to accommodate an appropriate scale.





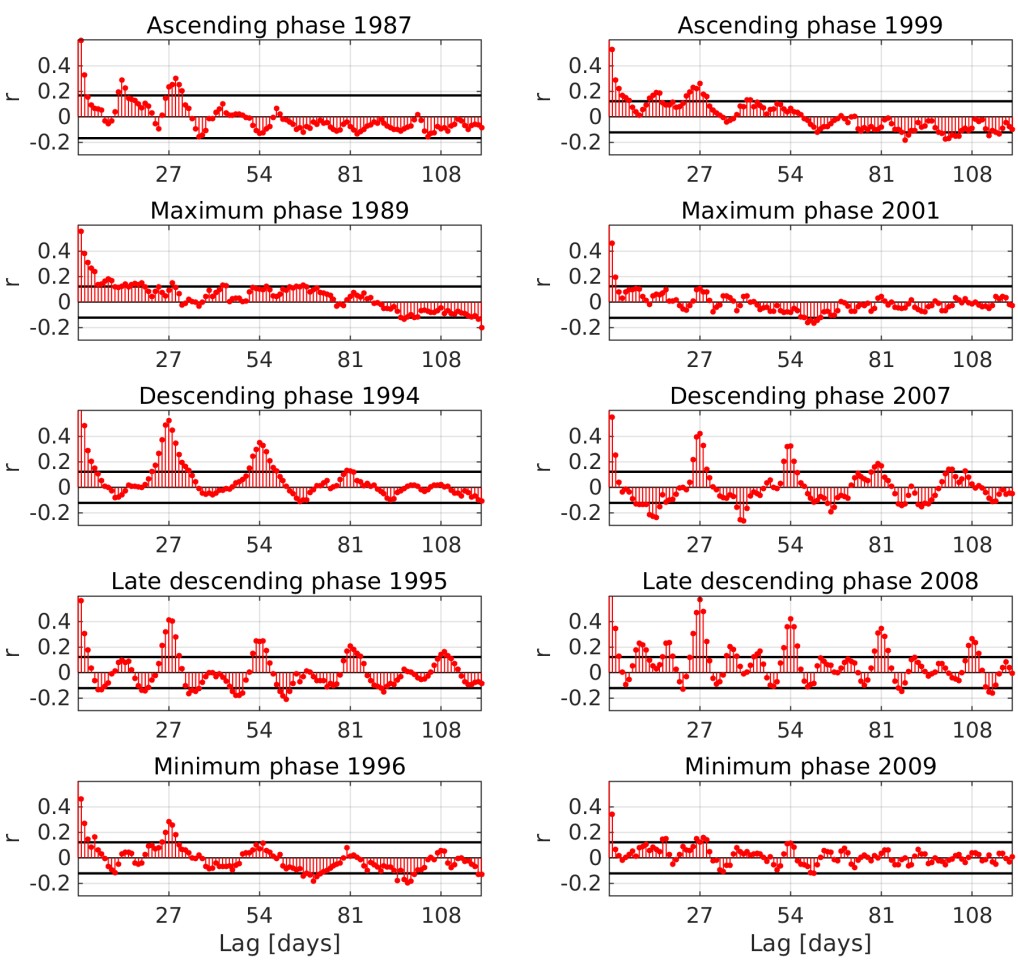

**Figure 3.** ACFs of Pc5 power for different phases of the SC 22 (left panels) and 23 (right panels) as a function of lag in days. The horizontal black lines are the critical values of correlation.

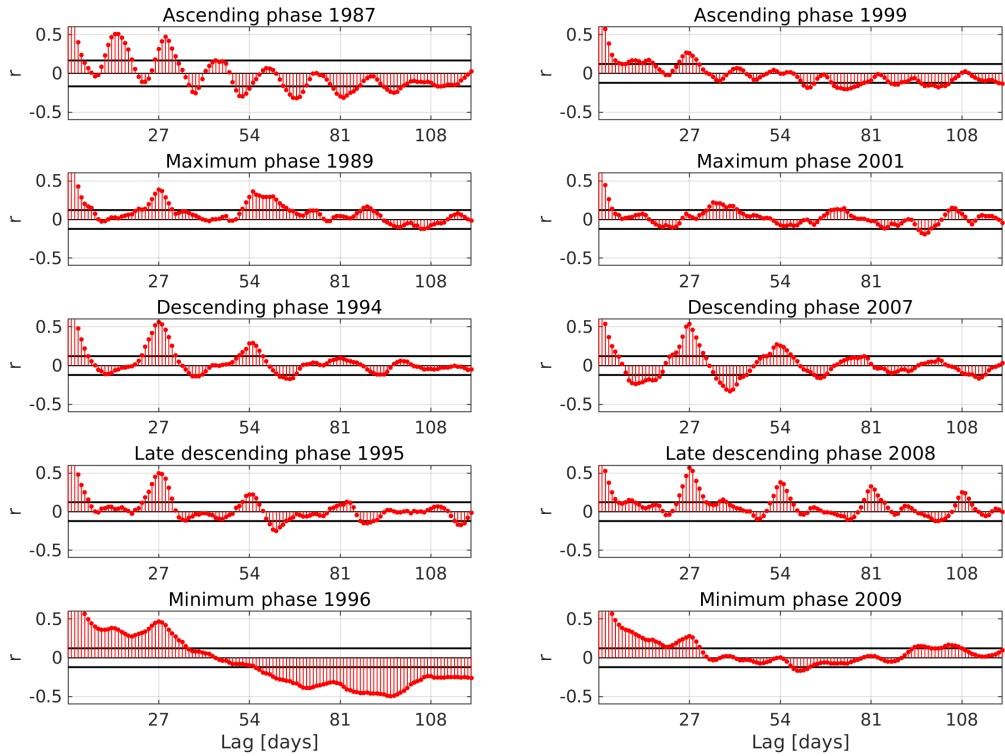

**Figure 4.** ACFs of electron fluence for different phases of the SC 22 (left panels) and 23 (right panels) as a function of lag in days. The horizontal black lines are the critical values of correlation.

In the ascending phase, Pc5 power shows two clear peaks that exceed the CVC around the 13 and 29 days lag in 1987. The $r$ values reach 0.3 in 1987. Similar peaks are discernible in 1999 though at a slightly different lag (e.g. peaking at 27 day lag instead of at 29 day lag in 1987). These peaks are also present in fluence. The peak around 29 day lag in SC 22 or the peak around 27 day lag in SC 23 approximates the Solar Synodic period ($T_{syn} \simeq 27.27$ days), which is due to the solar rotation and impinges a quasiperiodic 27-day variation upon many solar-terrestrial parameters (Poblet and Azpilicueta, 2018). The values over the CVC at $\sim 13$ days lag could be related to the 13-day periodicity that previous authors have found (e.g., Mursula and Zieger, 1996; Lam, 2004).

During the maximum phase, the peaks present in the ascending phase cannot be seen in Pc5 power as all the variations in 1989 and 2001 (Figure 3) are bounded by CVC, while in fluence the 27-day peak is still quite prominent in 1989 but less obvious in 2001 (Figure 4).

During the descending and late descending phases, the ACFs not only show the strongest values of correlation at 27 days lag, but also high values at $\sim 54$, $\sim 81$ and $\sim 108$, which are all multiples of 27. The value at day 27 reaches a maximum of 0.57





in 2008 for both Pc5 power and fluence. A peculiar characteristic that can be seen in Figure 3 in 2008 is that the correlation values exceeding CVC have a 9-day recurrence.

Although the peaks at multiples of 27 are quite sharp in the descending phase, the ACFs have smooth transitions between positive and negative values over the course of 27 days, suggesting that the Solar rotation generates a 27-day variation with a

sinusoidal-like pattern in these years as seen in the smooth progression of the anti-correlation values near the $\sim 13$ days lag. On the contrary, in the late descending phase the transitions between the peaks at multiples of 27 differs in both parameters. In fluence, we can deduce that the 27-day variation acts more like a spike owing to the flat correlation values in between solar rotations whereas in Pc5 power the lower harmonic with a period of $\sim 9$ days in 2008 is evident.

In the minimum phase, ACFs in fluence exhibit continuously moderate correlation values above CVC between lags 0 and

$\sim 30$ and negative trend thereafter with a reversal at large lag days between 1996 and 2009. For Pc5 power during the minimum phase, 1996 shows the persistent 27-day peak, which is present during other phases as mentioned above, and that peak is also present in 2009, though at a lower $r$.

### 3.1.2   A synopsis of the periodicities of Pc5 power and electron fluence over two solar cycles

The analysis developed in Section 3.1.1 provides a partial view of the periodicities as it only relates to specific years in

different SC phases. In this Section, we present the ACFs of each year of SC 22 and SC 23 together to trace the evolution of the periodicities throughout the two SCs. They are illustrated in two dimensional plots shown in Figure 5, which displays a synopsis of the periodicities of Pc5 power and electron fluence for the entire two solar cycles. The abscissa axis corresponds to the years and the ordinate axis to the lags (between 0 and 120 days). In the top-left(top-right) panel are the Pc5 power(fluence) ACFs where the neighboring values have been interpolated to visualize the trends better. In the bottom panels, the plots are

repeated for clarity with values of $|r|$ not exceeding $|CVC|$ shown as white bins.

From the almost continuous horizontal line of high correlation values centered at 27 days lag in all the panels, we can infer that the 27-day periodicity is the most prominent regular periodicity detected in Pc5 daily power as well as in fluence. In fact, all years except 1988 (ascending phase), 1998 (ascending phase) and 2001 (maximum phase) have values above CVC around the 27-day lag. 1994, 1995, 2006, 2007 and 2008 exhibit the strongest 27-day recurrence pattern with the highest correlation

values. All these years belong to the descending or late descending phase. The enhancements at multiples of 27 are also very clear. As the bottom panels show, they are dominant in the descending phase and absent in the ascending and maximum phase for both parameters.

The present understanding of the effects of 27-day variation that the solar rotation generates in the geospace environment can be used in the interpretation of our results. The regions known as Corotating Interaction Regions (CIRs) (Tsurutani et al.,

2006) are particularly important since they produce recurrent disturbances in geomagnetic activity as well as in other geospace phenomena. CIRs are formed when solar wind high-speed streams emanating from coronal holes into the interplanetary space catch the slow-speed streams, creating regions of enhanced density and magnetic fields.

During the declining phase of the SC, CIRs are particularly prominent as a result of the expansion of CHs to lower latitudes, generating a well developed sector structure in the heliospheric magnetic field. In this SC phase the ACFs of Pc5 power and

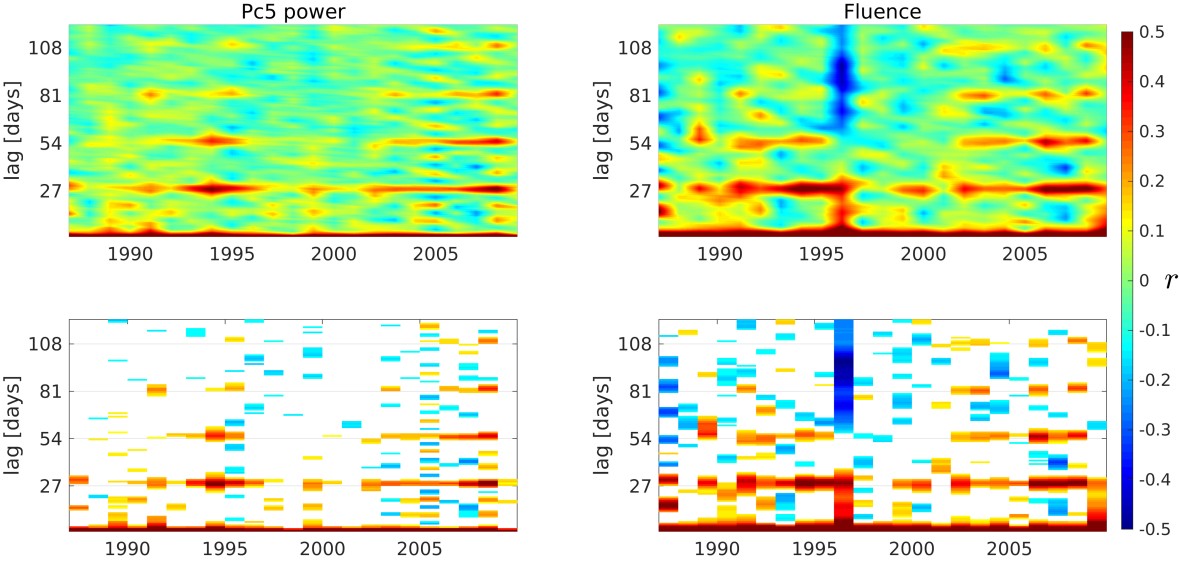

**Figure 5.** Two dimensional plots of ACFs of every year. In the abscissa axis is the year and in the ordinate axis is the lag (between 0 and 120 days). The top-left(top-right) panel shows the Pc5 power(fluence) ACFs with the neighboring values interpolated. White bins in the bottom panels belong to $|r|$ values that do not exceed $|\mathrm{CVC}|$.

fluence show the strongest values of correlation at 27-day lag as well as the clearest 27-day periodicity that repeats for several solar rotations. However, the fact that the ACFs peaks above CVC occur not only during the descending phase but also during other phases suggests that the 27-day variation in Pc5 power and fluence could also be due to smaller irregularities, other than CHs, capable of persisting for more than a solar rotation in the corona.

The peaks with a 9-day period seen in 2008 for Pc5 power (shown clearly in Figure 3), can also be seen in 2004 and 2005 as observed in the bottom-left panel of Figure 5. All of these years belong to the declining and late declining phases of SC23. On the contrary, only 2005 shows this periodicity clearly in fluence.

There are some previous reports of the 9-day recurrence in solar variables. For example, Ram et al. (2010) developed a comprehensive analysis of the solar rotation period and its subharmonics in the fractional area that CHs occupy at a fixed

region of the Sun and also in the solar wind velocity. They found that both parameters exhibit subharmonics with a period of 9 days during the declining and minimum phase of SC 23. Also, Temmer et al. (2007) and Lei et al. (2008) studied the prominent 9-day periodicity in the solar wind velocity on 2005 probing that it was caused by a triad of CHs separated by $\sim 120°$ in heliographic longitude that were active for several rotations. So the 9-day periodicity that we find in Pc5 power and fluence seems to be supported by prior investigations.

Finally, note that 1996 in fluence shows a different behavior than all the other years. This is evident when looking at this particular year in Figures 2 and 4.

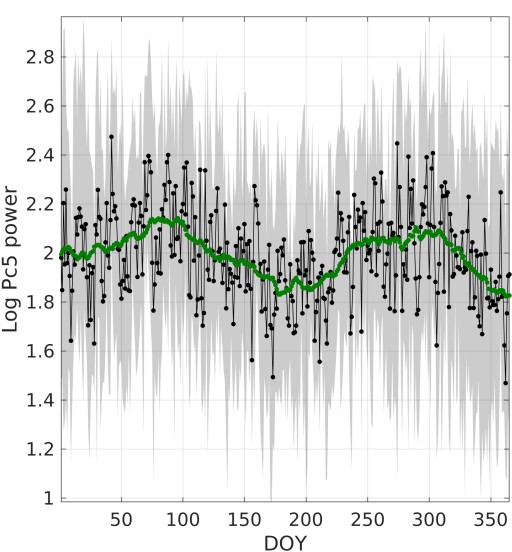
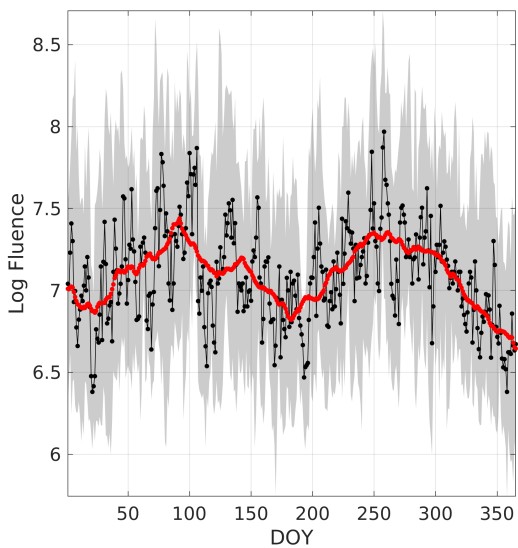

**Figure 6.** Superposed epoch analysis of the logarithmic values of Pc5 power (left panel) and fluence (right panel). The median and quartiles are illustrated as a black curve, lower limit and upper limit of the gray band respectively. The zero epoch is the first DOY and the green and red curve are the 30-day running average of the median curve (Pc5$_{SAV}$ and Fl$_{SAV}$).

## 4 Semiannual Variation (SAV)

In order to investigate the SAV in Pc5 power and electron fluence we performed a superposed epoch analysis to the logarithmic daily values of both parameters using the entire suite of two solar cycles data. The zero epoch was simply the first DOY and we calculated the median for each DOY from 1 to DOY 365 (the extra day corresponding to leap years was not used due to its negligible effect on the results). Owing to the length of the observations, there are $\sim 23$ values (corresponding to about 23 years) for each DOY to use in the calculation of the median. The results can be seen in Figure 6 which shows the superposed curve as black lines on the left and right panel corresponding to Pc5 power and electron fluence respectively. We chose the median over the mean for the superposition, since it is not skewed so much by extremely large or small values, and so it may give a better approximation to the "typical" value for each DOY. The upper and lower limits of the gray band mark the quartiles.

The 30-day running average of the curves with the median is also added in the figure (green line for Pc5 power and red line for electron fluence) and will be referred to as Pc5$_{SAV}$ and Fl$_{SAV}$. The 30-day moving average serves to diminish the strong 27-day variation since this is the most prevalent periodicity in both Pc5 power and fluence values, as shown in Section 3.1. Pc5$_{SAV}$ and Fl$_{SAV}$ demonstrate a clear SAV with maxima around the equinoxes and minima near solstices. Although not as clear, the SAV pattern can also be seen in the curves associated with the median and quartiles. The peak-to-peak variation of Fl$_{SAV}$ is of one order of magnitude approximately, and of $\sim 0.5$ orders of magnitude for Pc5$_{SAV}$. There are differences as well as similarities in the SAV of Pc5 power and fluence, and they will be discussed in Sections 4.1 and 4.2 below. Those

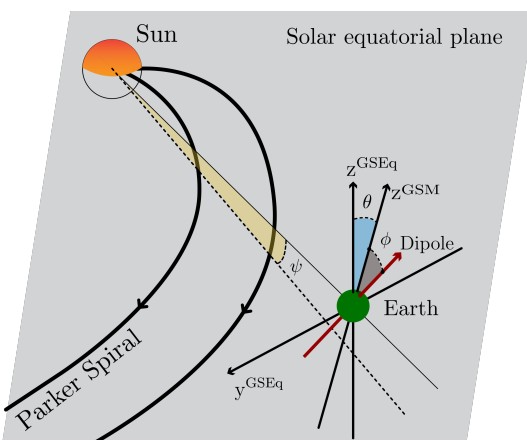

**Figure 7.** $\phi$, $\theta$ and $\psi$ in the Sun-Earth environment (read text for details). Parker spirals lie approximately in the solar equatorial plane that is shown in gray color. GSEq and GSM are the Geocentric Solar Equatorial and Geocentric Solar Magnetospheric coordinate systems respectively.

sections will explore in more detail the phases and profiles of the SAV in both parameters, but more importantly they will be compared with the phases and profiles predicted by the three classical hypotheses (introduced in Section 1) so that the dominant mechanism can be ascertained.

## 4.1  Annual profiles

In this Section we compared the profiles of the angles that govern each SAV mechanism (introduced in Section 1) with the profiles of Pc5$_{SAV}$ and Fl$_{SAV}$. For the axial hypothesis we considered the daily values of the Earth's heliographic latitude ($\psi$). For the equinoctial hypothesis we used daily mean values of the angle delimited by z$^{GSM}$ and the Earth's dipolar axis denoted by $\phi$ that is equivalent to the magnetic solar declination (with the same annual variation). Finally, for the RM effect we took the daily mean values of the angle between the z$^{GSM}$ and z$^{GSEq}$ axes that is measured in the y-z plane of both coordinates systems

(GSM and GSEq), referred to as $\theta$. Figure 7 shows schematically these three angles in the Sun-Earth environment where the gray plane is the solar equatorial plane.

Figure 8 presents the annual profiles of $|\theta|$, $|\psi|$ and $|\phi|$. The $|\phi|$ scale is inverted in the Figure in order to adequately identify the semiannual pattern in the three angles. As we are using daily mean values, the high frequency oscillations due to diurnal variations of $\phi$ and $\theta$ vanish. The three curves present a different overall shape for the seasonal modulation. For example, the

equinoctial mechanism anticipates sharper maxima and broader minima than the axial and RM hypotheses. The maxima and minima of the three angles fall on different dates and have different variation ranges, being $\sim 23°$ for $|\phi|$, $\sim 26°$ for $|\theta|$ and $\sim 8°$ for $|\psi|$. Note that $|\theta|$ is defined considering the GSEq coordinate system and not the Geocentric Solar Ecliptic (GSE) as in some works (Lockwood et al., 2016). This causes $|\theta|$ to reach slightly different values at the maxima. Considering GSEq



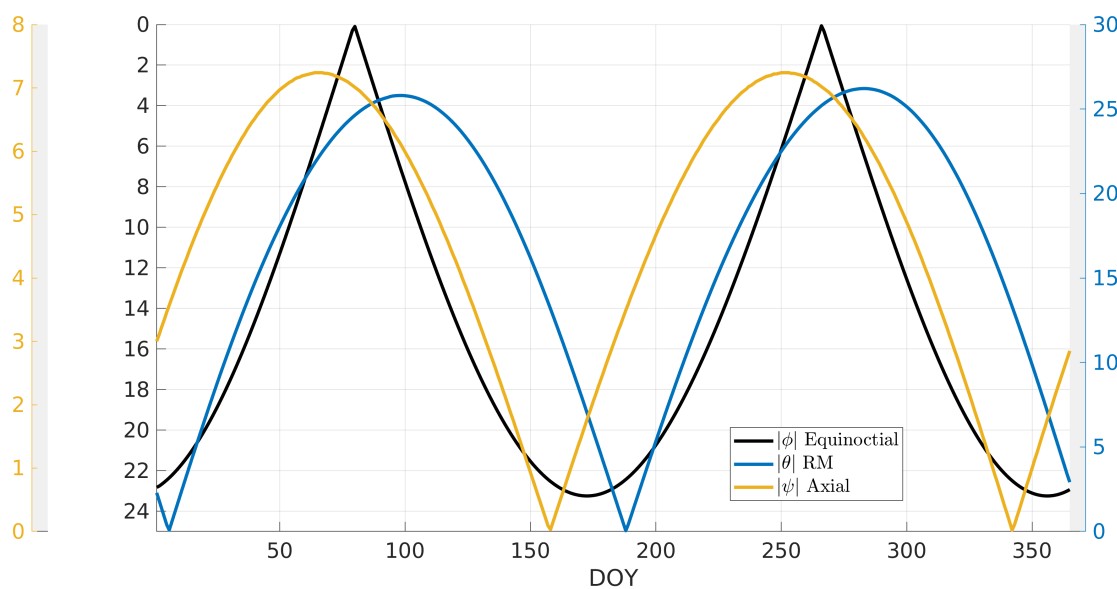

**Figure 8.** Absolute value of the angles that might control the SAV. $\phi$, $\theta$ and $\psi$ are associated to the Equinoctial, RM and Axial hypotheses respectively.

over GSE also delays the location of the maxima for several days and is more consistent with the original definition of the RM effect reported in Russell and McPherron (1973) (see for example Figure 4 on that paper).

To compare the shape of the angles with $FL_{SAV}$ and $Pc5_{SAV}$ we applied a 30-day running average to the curves in Figure 8. The results can be seen in Figure 9 where $FL_{SAV}$ and $Pc5_{SAV}$ are also illustrated at 3-day intervals. $FL_{SAV}$ and $Pc5_{SAV}$ 
follows better the Semiannual pattern between DOYs 180 and 365 approximately. In fact, between DOYs 200 and 250, $Pc5_{SAV}$ almost overlaps the smoothed inverted $|\phi|$ curve. Between DOYs 1 and $\sim$60, $FL_{SAV}$ and $Pc5_{SAV}$ reach higher values than the curves of the angles. In addition, as can also be seen in Figure 6, $FL_{SAV}$ shows sharper maxima than $Pc5_{SAV}$.

Some authors have used these three angles (or similar ones) in the past to test SAVs detected on magnetic indices. For example, Roosen (1966) used $ap$ index from 1932 to 1966 and determined that the annual pattern of the smoothed index 
presents greater similarity with the smoothed Equinoctial angle than with the smoothed Axial angle. Cliver et al. (2002) extended that comparison utilizing the 30-day smoothed patterns of the three angles and the $aa$ magnetic index from 1868 to 1998 obtaining high values of correlation with the smoothed $|\theta|$ but specially with the smoothed inverted $|\phi|$.

We calculated the correlation values between our observational curves ($FL_{SAV}$ and $Pc5_{SAV}$) and the smoothed angles and the results are summarized in Table 3. The equinoctial hypothesis seems to dominate the SAV in fluence since the correlation 
value between the smoothed $|\phi|$ and $FL_{SAV}$ profiles reaches the minimum value of $-0.87$, meaning that they anti-correlate very well. There is a lower fidelity of $FL_{SAV}$ with the RM profile (r $= 0.82$). As regards as $Pc5_{SAV}$, the profiles of both hypotheses (equinoctial and RM) show comparable correlation (anti-correlation) values, suggesting that the two mechanisms





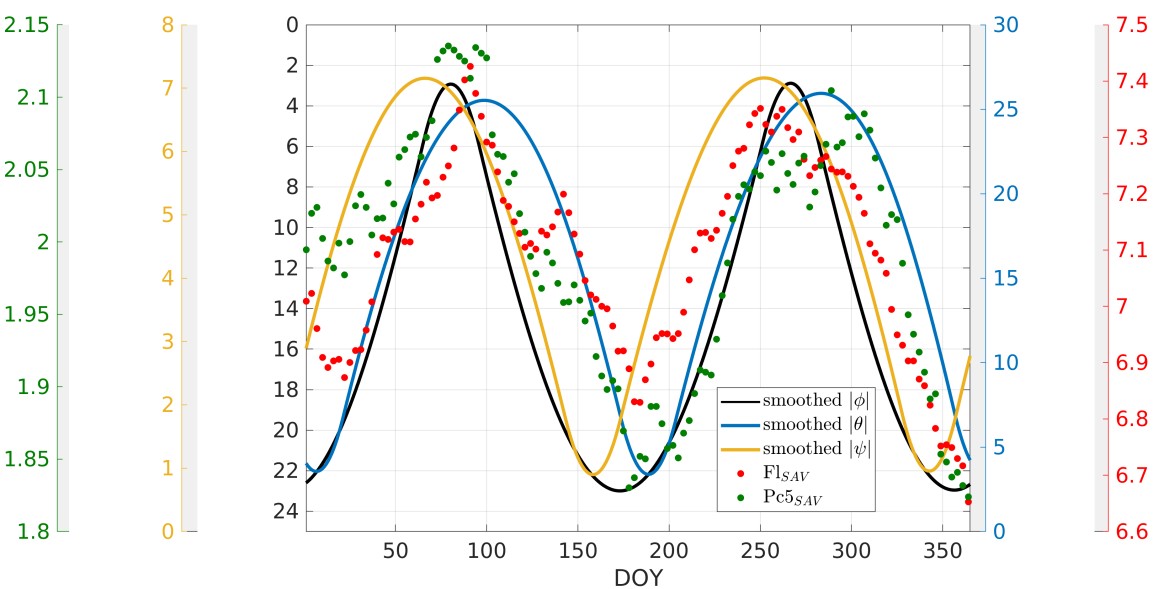

**Figure 9.** Smoothed absolute value of the angles that control the SAV. $\phi$, $\theta$ and $\psi$ are associated to the Equinoctial, RM and Axial hypotheses respectively. $\mathrm{FL}_{SAV}$ and $\mathrm{Pc5}_{SAV}$ have also been added, plotted at 3-day intervals.

|  | Smoothed angles | | |
|---|---|---|---|
|  | $|\phi|$ | $|\theta|$ | $|\psi|$ |
| $\mathrm{Fl}_{SAV}$ | $-0.87$ | $0.82$ | $0.69$ |
| $\mathrm{Pc5}_{SAV}$ | $-0.81$ | $0.80$ | $0.64$ |

**Table 3.** Correlation coefficients between the smoothed angles of the main semiannual hypotheses ( $|\phi|$, $|\theta|$ and $|\psi|$) and the observational curves ($\mathrm{Pc5}_{SAV}$ and $\mathrm{FL}_{SAV}$).

could play equally important roles in the generation of the SAV in Pc5 power. The profile of the Axial hypothesis presents the lowest agreement with both parameters (0.69 for $\mathrm{FL}_{SAV}$ and 0.64 for $\mathrm{Pc5}_{SAV}$).

### 4.1.1 Correlations with Functional dependencies of the angles

In principle, it should be possible to use the profiles of the three angles to determine which is the dominant mechanism, but
5   a better approximation may be achieved by considering functional dependencies of each angle. In this Section we evaluate functions of $\phi$ or $\theta$ proposed by different authors (Svalgaard, 1977; Perreault and Akasofu, 1978) in the past on studying the SAV in geomagnetic activity.



|  | S($\phi$) | Ak($\theta$) |
|---|---|---|
| Fl$_{SAV}$ | 0.88 | 0.80 |
| Pc5$_{SAV}$ | 0.83 | 0.79 |

**Table 4.** Correlation coefficients between functional dependencies of the angles (S($\phi$) and Ak($\theta$), read text for details) and observational curves (Pc5$_{SAV}$ and FL$_{SAV}$).

Svalgaard (1977) pointed out that the $am$ magnetic index can be fitted empirically using an expression for the magnetic field near a dipole, parameterized in terms of the controller angle of the equinoctial theory. The angular part of Svalgaard's function in terms of $\phi$ as defined in this work is S($\phi$) = $\left(1 + 3\cos^2(90° - \phi)\right)^{-2/3}$.

The angle $\theta$ of the RM hypothesis is considered in the "Akasofu" parameter (Perreault and Akasofu, 1978) that is usually utilized to characterize the energy brought by the SW to the magnetosphere. In addition to the SW and Interplanetary Magnetic field quantities involved in this proxy, the angular dependence is of the form Ak($\theta$) = $\sin^4(\theta/2)$. Finch and Lockwood (2007) determined that functions with this angular dependence are very successful on quantifying terrestrial disturbance levels on timescales of $\gtrsim$ 1 day.

We correlated S($\phi$) and Ak($\theta$) with FL$_{SAV}$ and Pc5$_{SAV}$ and the results are shown in Table 4. The correlation values of S($\phi$) are slightly better than to just using $|\phi|$ and the opposite occurs for Ak($\theta$) (see Table 3). However, all the correlation values are very similar to the ones obtained in Section 4.1 so no additional conclusions can be drawn.

## 4.2  Dates of maxima and minima

To continue the comparison with the three classical hypotheses, we determined the dates of maxima and minima of the SAV in fluence and Pc5 power and compared them with the corresponding dates of maxima and minima predicted by the three hypotheses.

First, we applied a non-linear least square fit with five parameters to the superposed median curves (black curves) of Figure 6. The following function was used:

$$f(t) = A^0 + A^{\mathrm{a}}\sin\left(\frac{2\pi}{365}t + \alpha^{\mathrm{a}}\right) + A^{\mathrm{sa}}\sin\left(\frac{4\pi}{365}t + \alpha^{\mathrm{sa}}\right);$$  (2)

with fixed annual and semiannual periodicities and the fitted parameters $A^0$, $A^{\mathrm{a}}$, $\alpha^{\mathrm{a}}$, $A^{\mathrm{sa}}$ and $\alpha^{\mathrm{sa}}$. $f(t)$ is plotted in Figure 10 as a green(red) curve in the left(right) panel that corresponds to the Pc5 power(fluence) fit. The other curves of Figure 10 are the median and quartiles as were presented in Figure 6.

Both fits follow the semiannual trend of the superposed median curves very well. In fact, the coefficient that modulates the amplitude of the annual variation is very low for both cases being $A^{\mathrm{a}} = -0.06$ for the fluence fit and $A^{\mathrm{a}} = 0.04$ for the Pc5 power fit. But in the semiannual term they are higher: $A^{\mathrm{sa}} = -0.23$ and $A^{\mathrm{sa}} = -0.12$ for fluence and Pc5 power respectively. An interesting characteristic that $f(t)$ reveals is that the minima on June/July and on December/January are not symmetric in





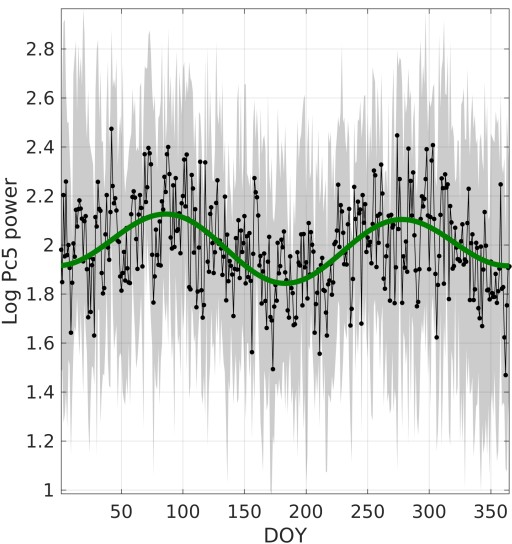
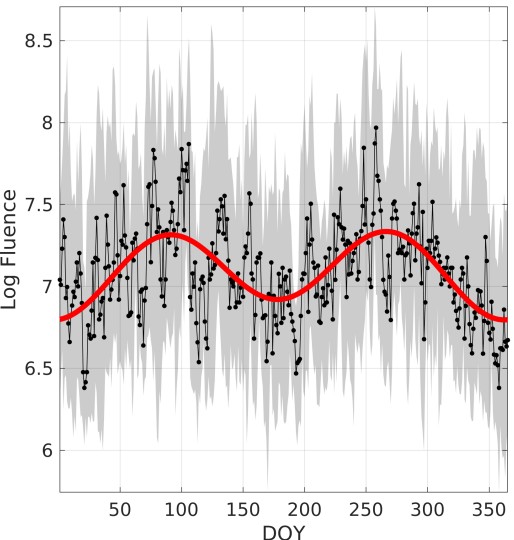

**Figure 10.** Superposed epoch analysis of the logarithmic daily values of Pc5 power (left panel) and fluence (right panel). The median and quartiles are illustrated as a black curve, lower limit and upper limit of the gray band respectively. The zero epoch is the first DOY. The green and red curve are fits of the median curve using $f(t)$ as in Equation 2. The parameters for the fit on the fluence curve are $(A^0, A^{\mathrm{a}}, \alpha^{\mathrm{a}}, A^{\mathrm{sa}}, \alpha^{\mathrm{sa}}) = (7.09, -0.06, 1.48, -0.23, 1.72)$ and for the Pc5 power fit $(A^0, A^{\mathrm{a}}, \alpha^{\mathrm{a}}, A^{\mathrm{sa}}, \alpha^{\mathrm{sa}}) = (2, 0.04, 1.28, -0.12, 1.61)$.

both fits. The minimum of $f(t)$ on June/July is lower than the minimum on December/January for Pc5 power and the opposite occurs with the fluence fit. On the contrary, $f(t)$ does not present this asymmetry for the maxima in both cases.

Once $f(t)$ was defined, we looked for the $t$ values that obey $\mathrm{d}f(t)/\mathrm{d}t = f'(t) = 0$ i.e. the times of maxima or minima of $f(t)$ referred to as $t_{\mathrm{max,min}}$, which will be used as the times of the maxima-minima of the SAV in fluence and Pc5 power. We applied
5   the so called "Newton–Raphson" method (Ypma, 1995) which is a classic method implemented to find zeros of a function.

We have $f'(t) = \frac{2\pi}{365} \left( A^{\mathrm{a}} \cos \left( \frac{2\pi}{365} t + \alpha^{\mathrm{a}} \right) + 2 A^{\mathrm{sa}} \cos \left( \frac{4\pi}{365} t + \alpha^{\mathrm{sa}} \right) \right)$. Expanding $f'(t)$ up to the linear term around an arbitrary value $\tilde{t}$ near $t_{\mathrm{max,min}}$ and setting $f'(\tilde{t}) \simeq 0$ we find

$$t \simeq \tilde{t} + \frac{A^{\mathrm{a}} \cos \left( \frac{2\pi}{365} \tilde{t} + \alpha^{\mathrm{a}} \right) + 2 A^{\mathrm{sa}} \cos \left( \frac{4\pi}{365} \tilde{t} + \alpha^{\mathrm{sa}} \right)}{\frac{2\pi}{365} \left( A^{\mathrm{a}} \sin \left( \frac{2\pi}{365} \tilde{t} + \alpha^{\mathrm{a}} \right) + 4 A^{\mathrm{sa}} \sin \left( \frac{4\pi}{365} \tilde{t} + \alpha^{\mathrm{sa}} \right) \right)}. \tag{3}$$

Calling $t = t_{n+1}$ and $\tilde{t} = t_n$, we iterated Equation 3 until $|t_n - t_{n+1}|$ was lower than a small value (a cut-off condition for
10   the iteration process), when $t_n$ becomes $t_{\mathrm{max,min}}$. The dates of the nominal equinoxes and solstices were utilized to initialize the iteration process.

The advantage of calculating $t_{\mathrm{max,min}}$ with this procedure is that Equation 3 also serves to estimate the errors in the determination of $t_{\mathrm{max,min}}$ because once $t_{\mathrm{max,min}}$ is determined, we can interpret Equation 3 as having $t$ expressed as a function of the parameters for values near $t_{\mathrm{max,min}}$, i.e $t = F(A^0, A^{\mathrm{a}}, \alpha^{\mathrm{a}}, A^{\mathrm{sa}}, \alpha^{\mathrm{sa}})$. Then, error propagation can be used in the determination of





|  | March/April maximum | June/July minimum | September/October maximum | December/January minimum |
|---|---|---|---|---|
| **Theoretical dates** | | | | |
| Equinoctial | 21 March | 22 June | 23 September | 22 December |
| Russell & McPherron | 7 April | 7 July | 11 October | 6 January |
| Axial | 7 March | 7 June | 9 September | 8 December |
| **Observed dates** | | | | |
| Fluence fit | 31 March ($\pm5.2$) | 26 June ($\pm5.5$) | 22 September ($\pm5.1$) | 27 December ($\pm5.0$) |
| Pc5 fit | 26 March ($\pm7.0$) | 1 July ($\pm6.6$) | 4 October ($\pm7.4$) | 28 December ($\pm7.7$) |
| **Correspondence of observed dates with theoretical dates** | | | | |
| Fluence fit | none | Equinoctial | Equinoctial | Equinoctial |
| Pc5 fit | Equinoctial | RM | RM | Equinoctial |

**Table 5.** Dates of maxima and minima for $|\phi|$, $|\theta|$ and $|\psi|$ and for the fits ($f(t)$) of the superposed median curve of Pc5 power and fluence.

$t$ with $F$. The maxima and minima dates ($t_{\mathrm{max,min}}$) with their uncertainty interval $2\sigma_t$ are shown in table 5. The table also shows the dates of maxima and minima predicted by the three mechanisms and which one of them falls into the uncertainty interval.

The best prediction of the SAV minima in fluence is given by the Equinoctial hypothesis. This mechanism is also the best one in estimating the September maximum with just one day of difference between the observed and predicted date. However, the three mechanisms fail to predict the March maximum in fluence that falls between the Equinoctial and RM predictions. Note that if the peaks and valleys times expected for the equinoctial mechanism are shifted forward 4 days as in (Kanekal et al., 2010), the fluence times of maxima/minima fall into the equinoctial uncertainty interval. This time shift was attributed by Li et al. (2001) and Kanekal et al. (2010) to finite solar wind speed ($\sim 440$ km s$^{-1}$).

For the SAV in Pc5 power it is not possible to find a dominant effect since the RM and the equinoctial theory give the best predictions for one maximum and one minimum but not both.

The results of this Section agree with the results found in the profiles analysis of Section 4.1. The equinoctial effect seems to be dominant in the generation of the SAV in fluence and both equinoctial and RM effects might be equally important for the SAV of Pc5 power.

## 5  Discussion

The previous sections have demonstrated a clear SAV in both parameters analyzed in this work. As a result of the length of the observations (two complete SCs of daily values) we were able to recover the background semiannual intensity variation in electron fluence and in Pc5 power. In the first case, this variation can be seen clearly in the red curve of the right-hand side panel of Figure 6 (Fl$_{SAV}$). Fl$_{SAV}$ reaches $\sim 7.5$ near equinoxes and $\sim 6.5$ near solstices that is equivalent to a difference





of one order of magnitude approximately. This means that there is a higher probability of internal charging on satellites near equinoxes then being more plausible for them to suffer operational anomalies. It also illustrates the way that the SAV influences space-based technologies.

In the study of the dominant effects, we found that the Equinoctial mechanism is dominant in the SAV of fluence and both

the Equinoctial mechanism and the RM effect play equally relevant roles in the SAV of Pc5 pulsations. These conclusions are reached by all the correlation values calculated in Sections 4.1, 4.1.1 and 4.2 in which there were analyzed the angle profiles of the three mechanisms, functional dependencies of the angles and also the location of maxima and minima dates that the mechanisms predict.

These results differ from previous ones reported in (Kanekal et al., 2010). Analyzing SAMPEX electron flux data, they found

a more prominent role for the RM effect. However, they considered fluxes in the heart of the outer radiation belt (L $\simeq$ 4) to evaluate the leading mechanism and not at GEO. So it is possible that different mechanisms may control the SAV in relativistic electrons in different regions of the magnetosphere. Another reason why we obtain different results could be that in (Kanekal et al., 2010) they used 10 years of daily values (from 1993 to 2002) which are less than half of the measurements processed in this work. Longer time spans of the data make the statistics more representative. In the case of the SAV of Pc5 pulsations, we

were not able to find prior reports studying the controller mechanism to compare with our findings.

It has been demonstrated before the potential of Pc5 power to predict electron fluence enhancements of individual events. In addition to individual events, we have shown that Pc5 power intensity is modulated with a semiannual pattern suggesting that this could be the origin of the SAV in electron fluence. Nevertheless, other elements must be evaluated to confirm this thesis so current works are being carried out in this direction by the authors.

Finally, other regular variations have also been studied in this work by means of the ACFs calculation. The main periodicities displayed by Pc5 power and fluence were tracked year by year along two complete 11-year SCs demonstrating that the 27-day period can be observed in every phase of the SC. And this period is most prominent during the declining phase when high correlations at multiples of 27 were also observed. On the contrary, the 27-day period is less recognizable in the ascending and maximum phase.

**6  Conclusions**

To summarize, this study demonstrates that Pc5 ULF waves and relativistic electrons both vary with multiple timescales due to the intrinsic periods of the Sun's dynamics, periods of the Earth's dynamics, and also those periodicities that result from considering the Sun-Earth system as a whole. In the 11-year solar cycle variation, this work affirms enhanced electron levels during the declining phase of a solar cycle, as previously reported in other studies. The 27-day periodicity of electrons presented

in this study is related to the recurrence of high-speed solar wind streams due to solar rotation. In explaining the SAV of electrons and Pc5 power by the three classical mechanisms, we have determined the most plausible ones to account for the observations. Similar SAV mechanisms as well as similar periodicities in both Pc5 power and electrons confirm that Pc5 ULF waves play an important role in energizing electrons, as attested to by other studies.





*Data availability.*

*Competing interests.* The authors declare that they have no conflict of interest.

*Acknowledgements.* The authors thank the producers of the GOES energetic electron data, which were downloaded from NOAA/NGDC. Pc5 wave power data that were derived from ground magnetic data recorded by NRCan's CANMOS are available from the authors upon
5   request (hlam@nrcan.gc.ca). The authors also thank the developers of the International Radiation Belt Environment Modeling (IRBEM) library that was used to calculate the theoretical angles used in this work (Bourdarie and O'Brien, 2009).





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
