# Peer review of "Semiannual variation of Pc5 ULF waves and relativistic electrons over two solar cycles of observations: comparison with predictions of the classical hypotheses"

_Annales Geophysicae, 2019_

## Referee Comment (RC1) · Anonymous Referee #1 · 3 Feb 2020

In their manuscript "Semiannual variations of Pc5 ULF waves and relativistic electrons over two solar cycles of observations: comparison with predictions of the classical hypotheses", Poblet et al. explore variations of power in the Pc5 frequency range observed by ground magnetometers and of relativistic electron fluence measured along the geosynchronous orbit. Through analysis of autocorrelation and superposed epoch analysis of data covering two solar cycles (22 and 23), the authors show variations in time scales ranging from days to months. Specifically, periodicities of approximately 9, 13 and 27 days, due to solar rotation have been identified in both relativistic elec-

tron fluence and Pc5 ULF wave power levels. Furthermore, an equinox maximum was observed in their seasonal variation, while lower level occurred around solstices throughout the year.

The presented results provide evidence pointing towards one order of magnitude higher electron fluence around solstices than equinoxes and 0.5 order of magnitude higher Pc5 ULF wave power around equinoxes than solstices. However, on the contrary to a previous publication by Lam (2011) that offered the starting point for this study, diurnal variation has not been considered even though it is expected to be of an order of magnitude in the electron flux measured along the geosynchronous orbit. If the author can address this concern such that their conclusions are clearly supported by the data presented and can improve the placement of this work in the context of previous literature, then this manuscript could become a valuable addition to the existing literature. Specifically, I could recommend this manuscript for publication in Annales Geophysicae subject to the specific points detailed below:

There are minor issues with English language use and several typographical errors.

For example, in line 1 and 5, acronyms such as ULF and GOES as well as NOAA on page 3, YKC, PBQ, BLC and CBB on page 4 should be expanded at first mention with the acronym provided in parenthesis after the acronym expansion.

Further down, in line 7, "though not present in all years" is followed by "are seen in some years" that essentially says the again the same thing already said.

On the next page 2, in line 9, the work of Summers and Ma (2000) is cited among the references for acceleration mechanisms of electrons in which Pc5 ULF waves have a key role to play. In parenthesis, however, it reads "Summers and yu Ma (2000)".

In line 14, the focus of the manuscript is introduced, namely variations in Pc5 ULF wave

power observed on the ground throughout the two previous solar cycles. Specifically, it reads "ground-based Pc5 magnetic pulsations, which are a manifestations of Pc5 ULF waves", contrary to the terminology widely employed today, which it is described in Section 1.1. of the following publication:

- McPherron (2005), Magnetic pulsations: Their sources and relation to solar wind and geomagnetic activity, Surveys in Geophysics, doi: 10.1007/s10712-005-1758-7

In Sections 2.1 and 2.2, the source of data used in this study is briefly described as well as the rationale behind their choice with key information missing. Although this manuscript presents the continuation of a previous study by Lam (2017), it has been submitted to published as a separate paper and should therefore stand on it own. Readers should not need to search for the publication of Lam (2017) to retrieve essential information about the data used to derive the presented results.

The choice of including measurements of Pc5 ULF wave power from the nightside magnetosphere along with those from the dayside magnetosphere and using electron fluence measurements from GOES satellites without considering the asymmetry in the dayside and nightside magnetosphere puzzles me as it seems inadequate to support the main conclusion of the study

Owing to the asymmetry of the magnetic field between the nightside and the dayside magnetosphere, satellites in almost circular orbits collect measurements from different (inner and outer) regions of the radiation belt. It is, therefore, difficult to separate temporal changes in the electron flux/fluence from changes due to the orbital motion of satellites.

Differences in measurements of electron flux/fluence along the satellite orbit could, however, be eliminated if they could be mapped at the same point. O'Brien et al. (2001) demonstrated a technique called Statistical Asynchronous Regression, which

determines the relationship between two time-varying quantities, without the need for simultaneous measurements of both quantities.

O'Brien et al. (2001) used this technique to map the flux round geosynchronous orbit to noon, as did Burin des Roziers and Li (2006) to map electron fluxes to other MLT. More recently, in Glauert et al. (2018), the technique has been employed to approximate the drift-averaged electron fluxes at a fixed L* from GOES data.

The publications referenced above are the following:

- O'Brien, T. P., Sornette, D., & McPherron, R. L. (2001). Statistical asynchronous regression: Determining the relationship between two quantities that are not measured simultaneously. Journal of Geophysical Research, 106(A7), 13,247–13,259. https://doi.org/10.1029/2000JA900193

- Burin des Roziers, E., & Li, X. (2006). Specification of >2 MeV geo-synchronous electrons based on solar wind measurements. Space Weather, 4, S06007. https://doi.org/10.1029/2005SW000177

- Glauert, S. A., Horne, R. B., & Meredith, N. P. (2018). A 30-year simulation of the outer electron radiation belt. Space Weather, 16, 1498–1522. https://doi.org/10.1029/2018SW001981

In lines 7 to 11, the choice to include data from the magnetosphere nightside is briefly explained. It would be noteworthy to add that a premidnight peak has been observed in GOES magnetic field data by Huang et al. (2010) and is likely the consequence of storm as well as substorm activity driven by tail processes, including substorm injections and dampened oscillatory flow in the plasma sheet. Lyons et al. (2002) has argued that ULF waves that strongly perturb the plasma sheet are a key component of tail dynamics during periods of enhanced convection. These ULF waves occasionally have amplitudes as large as plasma flow changes that occur in association with auroral

zone disturbances, such as substorms.

The publications referenced above are the following:

- Huang, C.-L, Spence, H. E., Singer, H. J., & Hughes, W. J. (2010). Modeling radiation belt radial diffusion in ULF wave fields: 1. Quantifying ULF wave power at geosynchronous orbit in observations and in global MHD model. Journal of Geophysical Research, 115, A06215. https://doi.org/10.1029/2009JA014917

- Lyons, L. R., Zesta, E., Xu, Y., Sanchez, E. R., Samson, J. C., Reeves, G. D., Ruohoniemi, J. M. & Sigwarth, J. B. (2002). Auroral poleward boundary intensifications and tail bursty flows: A manifestation of a large-scale ULF oscillation? Journal of Geophysical Research, 107(A11), 1352. https://doi.org/10.1029/2001JA000242

In lines 17 and 18, it would be more appropriate to read "horizontal axis" and "vertical axis" as the terms "abscissa" and "ordinate" are usually used to define the location of points in two-dimensional rectangular space.

In lines 15 and 16, the authors note that, during 1996, relativistic electron fluence shows a different trend in Figures 2 and 4. However, how this is different from relativistic electron fluence observed during the remaining time series analysed has not been described.

In line 4, it is not clear to me and perhaps the reader why the choice of displaying relativistic electron fluence and Pc5 ULF wave power has been selected to be displayed at intervals of three days. Would the choice of a longer or shorter intervals make a difference in the variation observed through the year?

In line 9, could the cut-off value in the condition |tn − tn+1| < "small value" checked before every iteration be provided?

In lines 16 to 19, the authors suggest that increases in Pc5 ULF wave power has been linked to relativistic electron fluence enhancements during individual events. However, I could not understand from the context whether geomagnetic storms are meant by individual events. In addition, references to such studies have not been provided.

The relationship with solar wind speed could also be discussed at this point along with seasonal variations in relativistic electron fluence and Pc5 ULF wave power. In the past, Lukianova et al. (2016) had looked into variations of solar wind speed over several solar cycles over the last 100 years.

Several studies have suggested that the solar wind speed is a dominant driver of relativistic electron fluxes in the outer radiation belt (e.g. Kellerman & Shprits, 2012, Paulikas & Blake, 1979). Furthemore, enhanced Pc5 ULF wave activity has associated with higher solar wind flow speed in the recovery phase of storms leading to enhanced electron fluxes (e.g. Georgiou et al., 2018, Mann et al., 2004).

The publication referenced above are the following:

- Lukianova, R., L. Holappa, & Mursula, K. (2017). Centennial evolution of monthly solar wind speeds: Fastest monthly solar wind speeds from long-duration coronal holes, Journal of Geophysical Research, 122, 2740–2747, https://doi.org/2016JA023683

- Kellerman, A. C., & Shprits, Y. Y. (2012), On the influence of solar wind conditions on the outer-electron radiation belt. Journal of Geophysical Research, 117, A05217. https://doi.org/10.1029/2011JA017253

- Paulikas, G. A., & Blake, J. B. (1979). Effects of the solar wind on magnetospheric dynamics: Energetic electrons at geosynchronous orbit, in Quantitative Modeling of Magnetospheric Processes. Geophysical Monograph Series, 21, 180–202

- Georgiou, M., Daglis, I. A., Rae, I. J., Zesta, E., Sibeck, D. G., Mann, I. R., Balasis, G., & Tsinganos, K. (2018). Ultra-low frequency waves as an intermediary for solar wind energy input into the radiation belts. Journal of Geophysical Research: Space Physics, 123, 10,090–10,108. https://doi.org/10.1029/2018JA025355

- Mann, I. R., O' Brien, T. P., & Milling, D. K. (2004), Correlations between ULF wave power, solar wind speed and relativistic electron flux in the magnetosphere: Solar cycle dependence. Journal of Atmospheric and Solar-Terrestrial Physics, 66(2), 187–198 (already included in the manuscript references)

---

## Referee Comment (RC2) · Anonymous Referee #2 · 12 Apr 2020

In this study the authors aim at presenting a detailed study of the correlation between PC5 ULF waves and enhancements of MeV electrons at GEO orbit. The follow the first study from Lam et al. (2017), and provide evidences of annual and semi-annual variability over two consecutive solar cycles. Moreover, they present insights to identify the major origins of these variabilities. The study is well detailed and numerous aspects are discussed. However, even if the authors rely on the previous study from Lam et al. (2017), the new findings are not enough highlighted, and conclusions do not provide fully new assets. I would recommend this work for publication after a few major

revisions. I detail in the following these points. Major remarks: 1- In Lam et al. (2017) the correlation is computed between electron fluxes and PC5 pulsations. Even if it is not the point in this study, I am thinking if the authors could discuss more these correlations, in particular in section 4.1. Figure 9 could benefit from more detailed cross-correlation between fluence and PC5 waves. As mentioned in the title of the manuscript, the reader is waiting for more details on such correlation in my mind. 2- Moreover, it is compared here with only > 2Mev electrons fluences. Do the authors tried to use the lower energy channel (>650keV electrons)? This may also add some discussion on the energization induced by these waves as well as radial diffusion, as a function of energy, as it has been discussed in some previous studies (see for example Lejosne et al., 2013). 3- One last major remark is (maybe naïve), why do the authors only discuss the power of the PC5 waves? Wouldn't it be interesting to discuss the correlation with fluence and solar cycle according to their modes (toroidal or poloidal as they tend to induce different effects on electrons trapped at GEO orbit, and as their sources may differ)? Minor remarks: 1- Page 3, line 9 : I think yGSEq should be changed into zGSEq, isn't it? 2- In section 4.1, there is only a sub-paragraph 4.1.1, but no 4.1.2. Please clarify.

---

## Author Comment (AC1) · 5 May 2020

*We would like to thank the reviewer for valuable comments. They have been perused carefully and responses to all of them are shown below. Our feedback for each comment are in the corresponding "Response" in red italics.*

In their manuscript "Semiannual variations of Pc5 ULF waves and relativistic electrons over two solar cycles of observations: comparison with predictions of the classical hypotheses", Poblet et al. explore variations of power in the Pc5 frequency range observed by ground magnetometers and of relativistic electron fluence measured along the geosynchronous orbit. Through analysis of autocorrelation and superposed epoch analysis of data covering two solar cycles (22 and 23), the authors show variations in time scales ranging from days to months. Specifically, periodicities of approximately 9, 13 and 27 days, due to solar rotation have been identified in both relativistic electron fluence and Pc5 ULF wave power levels. Furthermore, an equinox maximum was observed in their seasonal variation, while lower level occurred around solstices throughout the year. The presented results provide evidence pointing towards one order of magnitude higher electron fluence around solstices than equinoxes and 0.5 order of magnitude higher Pc5 ULF wave power around equinoxes than solstices.

However, on the contrary to a previous publication by Lam (2011) that offered the starting point for this study, diurnal variation has not been considered even though it is expected to be of an order of magnitude in the electron flux measured along the geosynchronous orbit. If the author can address this concern such that their conclusions are clearly supported by the data presented and can improve the placement of this work in the context of previous literature, then this manuscript could become a valuable addition to the existing literature. Specifically, I could recommend this manuscript for publication in Annales Geophysicae subject to the specific points detailed below:

**Response:** *This work has been developed using daily values because the aim is to study regular variations, and specifically the Semiannual Variation in a daily scale. We are aware that relativistic electron fluxes at geosynchronous orbit and Pc5 ULF wave powers undergo diurnal variations, so the main question is what value can be taken as representative of the day for both quantities. We show below that Fluence and the sum of Pc5 ULF powers at all local hours can be used for this purpose. The study of diurnal variations in both parameters could reveal that the Semiannual Variation appears only in a specific local-time sector, but this is beyond the scope of this paper.*

*A discussion about seasonal variations in solar wind speed can be found at the end of this document.*

There are minor issues with English language use and several typographical errors. For example, in line 1 and 5, acronyms such as ULF and GOES as well as NOAA on page 3, YKC, PBQ, BLC and CBB on page 4 should be expanded at first mention with the acronym provided in parenthesis after the acronym expansion. Further down, in line 7, "though not present in all years" is followed by "are seen in some years" that essentially says the again the same thing already said. On the next page 2, in line 9, the work of Summers and Ma (2000) is cited among the references for acceleration mechanisms of electrons in which Pc5 ULF waves have a key role to play. In parenthesis, however, it reads "Summers and yu Ma (2000)".

*Response: Thanks for this comment. All the acronyms will be expanded and the typographical errors will be corrected.*

In line 14, the focus of the manuscript is introduced, namely variations in Pc5 ULF wave power observed on the ground throughout the two previous solar cycles. Specifically, it reads "ground-based Pc5 magnetic pulsations, which are a manifestations of Pc5 ULF waves", contrary to the terminology widely employed today, which it is described in Section 1.1. of the following publication:
- McPherron (2005), Magnetic pulsations: Their sources and relation to solar wind and geomagnetic activity, Surveys in Geophysics, doi: 10.1007/s10712-005-1758-7

*Response: Thanks for this comment, we will change the terminology to "Pc5 ULF wave power".*

In Sections 2.1 and 2.2, the source of data used in this study is briefly described as well as the rationale behind their choice with key information missing. Although this manuscript presents the continuation of a previous study by Lam (2017), it has been submitted to published as a separate paper and should therefore stand on it own. Readers should not need to search for the publication of Lam (2017) to retrieve essential information about the data used to derive the presented results.

*Response: Thank you. An explanation of the data will be added in the manuscript as well.*

The choice of including measurements of Pc5 ULF wave power from the nightside magnetosphere along with those from the dayside magnetosphere and using electron fluence measurements from GOES satellites without considering the asymmetry in the dayside and nightside magnetosphere puzzles me as it seems inadequate to support the main conclusion of the study. Owing to the asymmetry of the magnetic field between the nightside and the dayside magnetosphere, satellites in almost circular orbits collect measurements from different (inner and outer) regions of the radiation belt. It is, therefore, difficult to separate temporal changes in the electron flux/fluence from changes due to the orbital motion of satellites.

Differences in measurements of electron flux/fluence along the satellite orbit could, however, be eliminated if they could be mapped at the same point. O'Brien et al. (2001) demonstrated a technique called Statistical Asynchronous Regression, which determines the relationship between two time-varying quantities, without the need for simultaneous measurements of both quantities. O'Brien et al. (2001) used this technique to map the flux round geosynchronous orbit to noon, as did Burin des Roziers and Li (2006) to map electron fluxes to other MLT. More recently, in Glauert et al. (2018), the technique has been employed to approximate the drift-averaged electron fluxes at a fixed L* from GOES data.

The publications referenced above are the following:
- O'Brien, T. P., Sornette, D., & McPherron, R. L. (2001). Statistical asynchronous regression: Determining the relationship between two quantities that are not measured simultaneously. Journal of Geophysical Research, 106(A7), 13,247–13,259. https://doi.org/10.1029/2000JA900193

- Burin des Roziers, E., & Li, X. (2006). Specification of >2 MeV geo-synchronous electrons based on solar wind measurements. Space Weather, 4, S06007. https://doi.org/10.1029/2005SW000177
- Glauert, S. A., Horne, R. B., & Meredith, N. P. (2018). A 30-year simulation of the outer electron radiation belt. Space Weather, 16, 1498–1522. https://doi.org/10.1029/2018SW001981

*Response: Due to asymmetric magnetic field, typical daily profiles of relativistic electron fluxes at geosynchronous orbit show maxima around noon and minima around midnight. These profiles can be seen in Figure 1 of (Su et al., 2014), which shows daily curves of >2 MeV electrons from GOES as a function of LT and Kp index activity. The curves in this figure show that typical daily pattern holds even when Kp reaches high values (disturbed times).*

*So if we want to work with a representative daily value to reveal the Semiannual Variation with maxima near Equinoxes and minima near Solstices we have several options. The first one, as the reviewer suggests, would be to apply ASR technique to map flux values at different MLTs to noon for example, and then calculate a mean daily value considering the flux at noon and all the mapped values. The averaged value should be very similar to the flux at noon.*

*Another option would be to simply take the flux at noon as representative of the day. This is the procedure followed by McPherron et al., (2009) to derive a Semiannual Variation with GOES 10 data covering Solar Cycle 22. The result is in the dashed line of Figure 4 in the mentioned paper.*

*A third option is to use Fluence as it is done in this work, that considers flux values at all LTs by means of the sum of the values.*

*However, since the orbit of the satellite is the same day by day, flux component that results from the orbit configuration will also be the same day by day. As a consequence, when the superposition is applied, this flux component will not affect the semiannual pattern that will be very similar in the three cases.*

*To prove this point, I have replicated Figure 4 in (McPherron et al., 2009) with Fluence data in our work, and the result is in Figure 1 at the end of this document. To improve the comparison, Figure 4 in (McPherron et al., 2009) has also been included at the end of this document.*

*Dashed line in Figure 1 shows the median of the Fluence ratio as a function of the DOY. The Fluence ratio is defined as the 27-day running average divided by 365-day running averages of the Fluence values in Solar Cycle 22.*

*In spite of the use of a different data set, the curve in Figure 1 is very similar to the one in the paper and shows clearly the Semiannual Variation.*

*It should be mentioned at this point that if we would like to study diurnal variations, UT variations of φ and θ introduced in the manuscript should also be considered. These are represented in Figure 2 at the end of this document. The proper quantity to use when working with daily values is the mean daily value of each angle as we have done in our work (Figure 8 of the manuscript).*

*The publications referenced above are the following:*

*-Su, Y.-J., J. M. Quinn, W. R. Johnston, J. P. McCollough, & Starks M. J. (2014). Specification of > 2 MeV electron flux as a function of local time and geomagnetic activity at geosynchronous orbit, Space Weather, 12, 470–486, doi:10.1002/2014SW001069.*
*-McPherron R.L., Baker D.N., & Crooker N.U. (2009). Role of the Russell–McPherron effect in the acceleration of relativistic electrons. Journal of Atmospheric and Solar-Terrestrial Physics, Volume 71, Issue 10-11, p. 1032-1044, doi:10.1016/j.jastp.2008.11.002.*

In lines 7 to 11, the choice to include data from the magnetosphere nightside is briefly explained. It would be noteworthy to add that a premidnight peak has been observed in GOES magnetic field data by Huang et al. (2010) and is likely the consequence of storm as well as substorm activity driven by tail processes, including substorm injections and dampened oscillatory flow in the plasma sheet. Lyons et al. (2002) has argued that ULF waves that strongly perturb the plasma sheet are a key component of tail dynamics during periods of enhanced convection. These ULF waves occasionally have amplitudes as large as plasma flow changes that occur in association with auroral zone disturbances, such as substorms. The publications referenced above are the following:

- Huang, C.-L, Spence, H. E., Singer, H. J., & Hughes, W. J. (2010). Modeling radiation belt radial diffusion in ULF wave fields: 1. Quantifying ULF wave power at geosynchronous orbit in observations and in global MHD model. Journal of Geophysical Research, 115, A06215. https://doi.org/10.1029/2009JA014917
- Lyons, L. R., Zesta, E., Xu, Y., Sanchez, E. R., Samson, J. C., Reeves, G. D., Ruohoniemi, J. M. & Sigwarth, J. B. (2002). Auroral poleward boundary intensifications and tail bursty flows: A manifestation of a large-scale ULF oscillation? Journal of Geophysical Research, 107(A11), 1352. https://doi.org/10.1029/2001JA000242

*Response: As a starting point, the objective was to evaluate periods and specifically study the Semiannual Variation considering powers at all local times together. Repeating the superposition and autocorrelation analyses to the powers of specific local times could give information about where the periods are produced. However, this does not mean that the main conclusions of the manuscript are invalid because the periods and the Semiannual intensity modulation are still clearly present in the daily values as they were used.*

*Moreover, excluding nigh-time powers should not change the results much because as it is pointed out in (Kozyreva et al., 2007), the correlation coefficient between ULF indices calculated for 00–24 and for 03–18 MLTs is very high at ~0.95, meaning that the substorm contribution to the daily Pc5 power would have been minor.*

*The publication referenced above is the following:*

*-Kozyreva, O., V. Pilipenko, M. J. Engebretson, K. Yumoto, J. Watermann, and N. Romanova (2007). In search of a new ULF wave index: Comparison of Pc5 power with dynamics of geostationary relativistic electrons, Planet. Space Sci., 55, 755–769*

In lines 17 and 18, it would be more appropriate to read "horizontal axis" and "vertical axis" as the terms "abscissa" and "ordinate" are usually used to define the location of points in two-dimensional rectangular space.

*Response: We will change the axes terminology. Thank you.*

In lines 15 and 16, the authors note that, during 1996, relativistic electron fluence shows a different trend in Figures 2 and 4. However, how this is different from relativistic electron fluence observed during the remaining time series analyzed has not been described.

*Response: We thank the reviewer for pointing this out. We indeed need to clarify why 1996 looks different. The different behavior of fluence values in 1996 is related to the distinct semiannual variation pattern of that year, as alluded to earlier in Figure 2. We will add this information after the sentence in lines 15 and 16 of Page 11.*

In line 4, it is not clear to me and perhaps the reader why the choice of displaying relativistic electron fluence and Pc5 ULF wave power has been selected to be displayed at intervals of three days. Would the choice of a longer or shorter intervals make a difference in the variation observed through the year?

*Response: No, the variation through the year is the same. Displaying the curves with a 3-day interval helps to improve the visualization since there are five time series plotted together in this figure. This information will be added to the Figure description.*

In line 9, could the cut-off value in the condition |tn – tn+1| < "small value" checked before every iteration be provided?

*Response: Yes, the value was 1E-14 that is reached after five iterations approximately. This information will be added to the text.*

In lines 16 to 19, the authors suggest that increases in Pc5 ULF wave power has been linked to relativistic electron fluence enhancements during individual events. However, I could not understand from the context whether geomagnetic storms are meant by individual events. In addition, references to such studies have not been provided.

*Response: By individual events we meant relativistic electron enhancements analyzed individually. Many enhancements take place during geomagnetic storms but they are not exclusively restricted to storm periods as it is pointed out in (Reeves et al., 2003). The paragraph will be rephrased to clarify this point.*

*The publication referenced above is the following:*

*-Reeves, G. D., McAdams, K. L., Friedel, R. H. W., and O'Brien, T. P. ( 2003). Acceleration and loss of relativistic electrons during geomagnetic storms, Geophys. Res. Lett., 30, 1529, doi:10.1029/2002GL016513.*

The relationship with solar wind speed could also be discussed at this point along with seasonal variations in relativistic electron fluence and Pc5 ULF wave power. In the past, Lukianova et al. (2016) had looked into variations of solar wind speed over several solar cycles over the last 100 years. Several studies have suggested that the solar wind speed is a dominant driver of relativistic electron fluxes in the outer radiation belt (e.g. Kellerman & Shprits, 2012,Paulikas & Blake, 1979). Furthemore, enhanced Pc5 ULF wave activity has associated with higher solar wind flow speed in the recovery phase of storms leading to enhanced electron fluxes (e.g. Georgiou et al., 2018, Mann et al., 2004). The publication referenced above are the following:

- Lukianova, R., L. Holappa, & Mursula, K. (2017). Centennial evolution of monthly solar wind speeds: Fastest monthly solar wind speeds from long-duration coronal holes, Journal of Geophysical Research, 122, 2740–2747, https://doi.org/2016JA023683
- Kellerman, A. C., & Shprits, Y. Y. (2012), On the influence of solar wind conditions on the outer-electron radiation belt. Journal of Geophysical Research, 117, A05217. https://doi.org/10.1029/2011JA017253
- Paulikas, G. A., & Blake, J. B. (1979). Effects of the solar wind on magnetospheric dynamics: Energetic electrons at geosynchronous orbit, in Quantitative Modeling of Magnetospheric Processes. Geophysical Monograph Series, 21, 180–202
- Georgiou, M., Daglis, I. A., Rae, I. J., Zesta, E., Sibeck, D. G., Mann, I. R., Balasis, G., & Tsinganos, K. (2018). Ultra-low frequency waves as an intermediary for solar wind energy input into the radiation belts. Journal of Geophysical Research: Space Physics, 123, 10,090–10,108. https://doi.org/10.1029/2018JA025355
- Mann, I. R., O' Brien, T. P., & Milling, D. K. (2004), Correlations between ULF wave power, solar wind speed and relativistic electron flux in the magnetosphere: Solar cycle dependence. Journal of Atmospheric and Solar-Terrestrial Physics, 66(2), 187–198 (already included in the manuscript references)

*Response: Thanks for this comment. A brief discussion of the relationship between electron fluxes, Pc5 ULF wave power and solar wind speed will be added. An interesting point is that solar wind speed does not present a recognizable semiannual pattern as electron fluxes and Pc5 ULF wave powers do. In fact, this is a strong argument to discard the Axial hypothesis. The reviewer may check Figure 4 in (McPherron et al., 2009) (that is at the end of this document) that shows a superposed epoch analysis for solar wind speed in which no seasonal pattern can be identified.*

*In (Lukianova et al., 2017) they calculate monthly linear regressions between DH (disturbed values of H geomagnetic component) and V (solar wind velocity) as: V = a DH + b. Then, they plot coefficients a and b (Figure 2) showing that a clear Semiannual Variation can be observed. This supports the idea that solar wind speed does not have any seasonal pattern in the following manner.*

*Since it is known that H component has a Semiannual Variation (see for example (Azpilicueta et al., 2012) ), if V would have this variation the coefficients of the fits should not show any seasonal pattern because the slope and intercept value would not vary much from month to month. So the fact that the slope and intercept value seasonally change is a consequence of a seasonality in DH and a lack of a seasonal behavior in V.*

*The publication referenced above is the following:*

-Azpilicueta, F., and Brunini, C. (2012), A different interpretation of the annual and semiannual anomalies on the magnetic activity over the Earth, J. Geophys. Res., 117, A08202, doi:10.1029/2012JA017893.

*Figure 1*

[Figure]

*Figure 2*

[Figure]

*Figure 4 in (McPherron et al., 2009)*

[Figure]

[Figure]

**Fig. 4.** The semiannual variation of the flux of relativistic electrons at synchronous orbit during sunspot cycle #22 is illustrated by the median of the flux ratio as a function of time relative to summer solstice (dashed black line). The flux ratio is defined as the 27-day running average time series of synchronous noon flux divided by 365-day running averages of this flux. The solar wind velocity does not show a similar modulation as demonstrated by the shaded band near a ratio of 1.0.

---

## Author Comment (AC2) · 5 May 2020

*We would like to thank the reviewer for valuable comments. They have been perused carefully and responses to all of them are shown below. Our feedback for each comment are in the corresponding "Response" in red italics.*

In this study the authors aim at presenting a detailed study of the correlation between PC5 ULF waves and enhancements of MeV electrons at GEO orbit. The follow the first study from Lam et al. (2017), and provide evidences of annual and semi-annual variability over two consecutive solar cycles. Moreover, they present insights to identify the major origins of these variabilities. The study is well detailed and numerous aspects are discussed. However, even if the authors rely on the previous study from Lam et al. (2017), the new findings are not enough highlighted, and conclusions do not provide fully new assets. I would recommend this work for publication after a few major revisions. I detail in the following these points.

*Response: Thanks for pointing out this. The main results can be summarized in four items as:*

1. *Relativistic electron Fluences present a clear Semiannual Variation. Logarithmic Fluences reach ~7.5 near equinoxes and ~6.5 near solstices, equivalent to a difference of approximately one order of magnitude. This means that there is a higher probability of internal charging on satellites near equinoxes then being more plausible for them to suffer operational anomalies.*
2. *Pc5 ULF wave powers also have a Semiannual pattern being ~0.5 orders of magnitude higher near equinoxes than near solstices.*
3. *Due to all correlations in Sections 4.1, 4.1.1 and 4.2, it can be inferred that the Equinoctial mechanism may be the dominant effect in the Semiannual Variation of Fluence and both Equinoctial and RM mechanisms would play equally relevant roles in the Semiannual Variation of Pc5 ULF wave powers.*
4. *The autocorrelation analyses served to track periods in both parameters along two 11-year solar cycles (SCs). The 27-day period can be observed in every phase of the SC being most prominent during the declining phase when high correlations at multiples and subharmonics of 27 were also observed. On the contrary, the 27-day period is less recognizable in the ascending and maximum phase.*

*These four points are put in context in Sections 3.1.2 and 5 where they are compared and discussed with results obtained in previous works.*

*We will modify the text so that the main results are clear for the reader.*

Major remarks:

1- In Lam et al. (2017) the correlation is computed between electron fluxes and PC5 pulsations. Even if it is not the point in this study, I am thinking if the authors could discuss more these correlations, in particular in section 4.1. Figure 9 could benefit from more detailed cross-correlation between fluence and PC5 waves. As mentioned in the title of the manuscript, the reader is waiting for more details on such correlation in my mind.

*Response: The title refers to the comparison of the Semiannual Variation in both parameters (Pc5 ULF waves and relativistic electron Fluences) with the three main theories (Axial, Equinoctial and RM) which reflects the objective of the study.*

*One has to be cautious in saying that the cause of Semiannual Variations in relativistic electrons is the Semiannual Variation in Pc5 ULF waves because there are many physical processes that can produce electrons at MeV energies in the magnetosphere. So we decided to limit the comparison between $Fl_{SAV}$ and $Pc5_{\_SAV}$ and focus on the comparison with the main theories.*

*However, a brief discussion of the results in (Lam, 2017) will be added since it could be valuable to highlight the main results of the manuscript.*

2- Moreover, it is compared here with only > 2Mev electrons fluences. Do the authors tried to use the lower energy channel (>650keV electrons)? This may also add some discussion on the energization induced by these waves as well as radial diffusion, as a function of energy, as it has been discussed in some previous studies (see for example Lejosne et al., 2013).

3- One last major remark is (maybe naïve), why do the authors only discuss the power of the PC5 waves? Wouldn't it be interesting to discuss the correlation with fluence and solar cycle according to their modes (toroidal or poloidal as they tend to induce different effects on electrons trapped at GEO orbit, and as their sources may differ)?

*Response: The suggestion of studying low energy electrons as well as poloidal and toroidal modes would certainly improve the knowledge of the influence of Pc5 ULF waves on magnetospheric electrons. We are considering to pursue such topics in our future studies.*

*However, we think the study of periods and Semiannual Variation in both sets of observations used in this manuscript is long enough and self sufficient to present it in a paper as it is.*

Minor remarks:

1- Page 3, line 9 : I think yGSEq should be changed into zGSEq, isn't it?

*Response: Thank you for this comment. It should be written zGSEq to be consistent with the angle θ used in Section 4.1. However, the use of yGSEq instead of zGSEq means just a 90° shift and the yearly variation with respect of zGSM maintains.*

2- In section 4.1, there is only a sub-paragraph 4.1.1, but no 4.1.2. Please clarify.

*Response: Thanks. 4.1.1 could be just 4.2 and then change the numeration of the following sections.*

---

## Author Response (AR2)

Author's Response:

- The minor suggestion of reviewer 1 has been followed. We added a brief discussion of similarities and differences between the shape of the observed Semiannual Variations ($FI_{SAV}$ and $Pc5_{SAV}$ ) and the predicted semiannual patterns of the three angles for DOYs between 1 and 179. The new paragraph can be found on page 13 line 261.